# Effects of Sodium Hexametaphosphate Addition on the Dispersion and Hydration of Pure Calcium Aluminate Cement

**DOI:** 10.3390/ma13225229

**Published:** 2020-11-19

**Authors:** Benjun Cheng, Can Yao, Jian Xiong, Xueyin Liu, Haijun Zhang, Shaowei Zhang

**Affiliations:** 1School of Energy Science and Engineering, Central South University, Changsha 410083, China; chbj666@csu.edu.cn (B.C.); can.yao@csu.edu.cn (C.Y.); xj1028@mail.csu.edu.cn (J.X.); 2College of Civil Engineering and Architecture, Quzhou University, Quzhou 324000, China; liuxueyin@cujb.edu.cn; 3The State Key Laboratory of Refractory and Metallurgy, Wuhan University of Science and Technology, Wuhan 430081, China; 4College of Engineering, Mathematics and Physical Science, Exeter University, Exeter EX4 4QF, UK

**Keywords:** pure calcium aluminate cement, sodium hexametaphosphate, adsorption, *ζ*-potential, rheological properties

## Abstract

The effects of sodium hexametaphosphate (SHMP) addition on the dispersion and hydration of calcium aluminate cement were investigated, and the relevant mechanisms discussed. The content of SHMP and the adsorption capacity of SHMP on the surface of cement particles were estimated using plasma adsorption spectroscopy and the residual concentration method. The rheological behavior of hydrate, *ζ*-potential value of cement particles, phase transformation and the microstructure of the samples were determined by coaxial cylinder rheometer, zeta probe, X-ray diffraction (XRD) and scanning electron microscopy (SEM). The results indicate that SHMP readily reacted with Ca^2+^, forming complexes [Ca_2_(PO_3_)_6_]^2−^ ions which were subsequently adsorbed onto the surfaces of cement particles. When the content of SHMP was 0.05%, the adsorption ratio reached 99%. However, it decreased to 89% upon further increasing the addition of SHMP to 0.4%. The complexes [Ca_2_(PO_3_)_6_]^2−^ adsorbed onto the surfaces of cement particles inhibited the concentration of Ca^2+^ and changed *ζ*-potential, resulting in enhanced electrostatic repulsive force between the cement particles and reduced viscosity of cement-water slurry. The experimental results indicate that the complexes [Ca_2_(PO_3_)_6_]^2−^ covering the surfaces of cement particles led to a delayed hydration reaction, i.e., they extended the hydration time of the cement particles, and that the optimal addition of SHMP was found to be about 0.2%.

## 1. Introduction

Superplasticizers around the world can be divided into several distinct types based on different chemicals; they have a similar function, that is, to reduce the water content of concrete without loss of workability [1,2,3]. Normally, most of these chemicals are high molecular weight organic compounds, some are synthetic and others are derived from natural products. The main water- reducing agents are based on naphthalene, melamine, lignosulfonate and other natural compounds, such as glucose, sucrose and hydroxylated polymers, and salts of organic hydroxycarboxylic acids, etc. [4,5,6]. Water reducer is usually a kind of surfactant, which belongs to anionic surfactant. After the superplasticizer dissolves into water, it will dissociate negative ions, which will be adsorbed on the surface of cement particles. When the adsorption reaches a certain degree, the cement particles show electrical properties. The charge between cement particles is the same. Under the action of electrostatic repulsion, the particles are dispersed, and a solvated water film is formed on the surface of the particles to reduce the surface tension of water, so that the fluid can penetrate into the solid particles and the dense mixture has enough water permeability. Due to the electrostatic repulsive force and the lubricating effect of the solvent water film, the cohesive structure (flocculation structure) between the cement particles is broken up, releasing the free water that it encloses. A small amount of superplasticizer can reduce the water content of cement, but it does not affect the fluidity or workability of cement [7,8]. On the other hand, because of the addition of water-reducing agent, the surface of cement particles forms an adsorption film, which affects the hydration speed of cement, makes the growth of cement stone crystal more perfect, reduces the capillary gap of water evaporation, and makes the network structure more compact, which improves the hardness and structural densification of cement mortar [9,10,11].

Fluidity is considered one of the basic properties of cement-based materials, because it plays an important role in determining the workability of construction and various properties of hardened materials [12]. Superplasticizers are usually added to improve the fluidity of cement slurry [13]. In recent decades, a variety of superplasticizers with different dispersing capacity have been developed and applied in cement-based materials. Among them, polycarboxylate (PCE), lignosulfonate, etc., superplasticizers are commonly used in Portland cement used in building materials [14,15,16]. However, the aluminate cement used as binder in refractory pouring materials generally uses sodium hexametaphosphate (SHMP), sodium tripolyphosphate (STP), etc., superplasticizers [17]. In the unshaped refractory industry, calcium aluminate cement (mainly composed of CA and CA_2_) is an excellent adhesive for the preparation of high-performance refractories because of its good early strength, high fire resistance and good wear resistance [18]. Sodium hexametaphosphate (SHMP) is commonly used as a superplasticizer for calcium aluminate cement (CAC) composite castables with a silica fume content of 2–8%; because of its long chain, it can ensure that the refractory castable can achieve the required fluidity and easy pouring construction performance, and promote the refractory casting construction lining to form a low porosity and high-mechanical-strength structure [19,20].

In recent years, the behavior and performance of SHMP in different systems has been studied by several researchers. Xu et al. [21] tried to develop a kind of chemically bonded refractory castable based on acid-based reaction between calcium aluminate cement (CAC) and polyphosphate-based reactants. It was found that the addition of sodium hexametaphosphate (SHMP) significantly improved the fluidity of the resulting binder, which was due to its good anti flocculation effect and sufficient hydration of CAC particles [22]. Ma and Brown used sodium phosphate to modify the hydration of CAC, thereby inhibiting the formation of metastable CAH_10_ and C_2_AH_8_, and avoiding subsequent conversion reactions to become stable C_3_AH_6_ [23,24,25]. Although, many scholars have done many studies on SHMP, only limited work has been done on the dispersing/retarding effect of SHMP on pure calcium aluminate cement. For example, the amount of SHMP adsorbed on cement particles is related to what factors, how to achieve the water reducing effect, how the cement hydration process depends on the amount of SHMP added and a series of related issues. The relevant mechanism still has not been fully understood. To address this, in the present work, *ζ*-potential, the adsorption amount of SHMP, the concentrations of P element and Ca^2+^ and rheological properties corresponding to different SHMP additions were examined, based on which, and the hydration behavior of SHMP investigated, the relevant mechanism underpinnings were discussed. The work would provide an insight into the effects of SHMP on the dispersion and water reduction in a PCAC (pure calcium aluminate cement)-bonded castable system.

## 2. Experimental

### 2.1. Raw Materials

Pure calcium aluminate cement (SECAR 71) was supplied by the Sinosteel Luoyang Institute of Refractories Research (Luoyang, China), the chemical composition of which is listed in Table 1. Analytical-grade SHMP was purchased from Luoyang Hexin Refractories Co., Ltd (Luoyang, China).

### 2.2. Sample Preparation and Characterization

#### 2.2.1. Adsorption of SHMP onto PCAC Cement

A total of 2.5 g PCAC was mixed with 50 mL SHMP solution with a concentration of 0%, 0.05%, 0.1%, 0.2%, and 0.4%, respectively, and stirred with a glass rod for 5 or 30 min. A total of 5 mL of the filtrate was taken, and its P element and Ca^2+^ concentrations were determined by plasma adsorption spectroscopy (ICAP 6000 SERIES, Cambridge, UK) and used to estimate the SHMP content in it. The amounts of SHMP adsorbed onto the surfaces of cement particles were evaluated by the residual concentration method [26].

#### 2.2.2. ζ-Potential of Cement Particles

ζ-potential values of cement particles were measured by zeta probe (ZetaProbe, LWL Development Limited, Hong Kong, China). A total of 10 g PCAC was mixed with 200 mL SHMP solution with a concentration of 0%, 0.05%, 0.1%, 0.2%, and 0.4%, respectively, and the suspension was stirred. After 5 min, the zeta potential of the cement particles was measured by a potentiometer [27]. During the experiment, the pH value of the system was around 11.2, decreasing slightly with the test time increasing, reaching the standard pH value (11–12) in the CAC system.

#### 2.2.3. Rheological Behavior of Cement-Water Slurry

PCAC was also mixed with water in a given weight ratio of 7/3, followed by the addition of respectively 0%, 0.05%, 0.1%, 0.2% and 0.4% SHMP. A rheometer (MCR301, Anton Paar, Styria, Austria) with a coaxial cylinder geometry (ST22-4V-40 system) was used to examine the rheological behavior of each slurry at 37 °C. (This experiment is carried out at room temperature, in summer.) For the static state flow test, the shear rate used was in the range from 0.1 s^−1^ to 1000 s^−1^.

#### 2.2.4. Hydration Behavior of PCAC

A total of 2 kg PCAC was mixed with, respectively, 0%, 0.05%, 0.1%, 0.2% and 0.4% SHMP, followed by the addition of appropriate amounts of water under stirring. The resultant wet mix was cast into a mould of 40 mm × 40 mm × 160 mm. The samples were demoulded after natural curing for 24 h in a thermostat at 37 °C (simulating normal summer temperature). Phase changes in the samples were analyzed by X-ray diffraction (XRD) (PANalytical, Empyrean, The Netherlands), and their microstructures were observed by using a scanning electron microscope (SEM) (Nova400NanoSEM, Amsterdam, The Netherlands).

## 3. Results and Discussion

### 3.1. Adsorption Behavior of SHMP

Based on the residual concentration method, the concentrations of P element determined by ICAP can be used to estimate the amounts of SHMP in the filtrate and adsorbed onto the surfaces of cement particles. At the beginning of the experiment, the cement particles sank to the bottom, and the adsorption amount of SHMP on its surface was very little, almost none. The concentration of p element was measured and compared after standing for 5 min and 30 min, and it was found that there was no change. Therefore, stirring was adopted to enhance its adsorption capacity. Figure 1 shows these two amounts as a function of total addition of SHMP and stirring time, revealing that the amounts of SHMP adsorbed onto the surfaces of cement particles increased almost linearly with the total addition of SHMP (Figure 1c,d). Figure 2 further illustrates the change of the adsorption ratio of SHMP (i.e., the ratio of the amount of SHMP adsorbed onto cement particles to the total addition of SHMP) with the total addition of SHMP and stirring time. When 0.05% SHMP was added, the adsorption ratio after 5 min stirring reached 99.6%. However, it decreased to 89.2% upon increasing the SHMP addition to 0.4%. Furthermore, comparison of Figure 2a,b reveals that, for a given addition of SHMP, the adsorption ratio after 30 min stirring was greater than that after 5 min stirring, indicating that the stirring assisted the adsorption of SHMP onto the cement particles, as the stirring time increased from 5 min to 30 min, the adsorption capacity was enhanced.

From Figure 1a,b, it also can be seen that only minor amounts of P element were detected in the filtrates. This might be due to one of the following reasons: (1) the chelation of SHMP with Ca^2+^ in the solution resulted in insoluble phosphates, and (2) the chelation of SHMP with Ca^2+^ in the solution resulted in soluble complexes which were adsorbed onto the surfaces of cement particles, and re-precipitated with cement particles. Previous studies [28,29,30] showed that the phosphates generated from the strong chelation of SHMP with Ca^2+^ were water soluble, so the first case could be ruled out. Therefore, it can be considered that the complexes formed from the chelation of SHMP with Ca^2+^ (see Reaction (1) in Section 3.2 below) were adsorbed onto the surfaces of cement particles, leaving only a little P element in the filtrates.

### 3.2. Ca^2+^ Concentration in Filtrate

Figure 3 illustrates the change of Ca^2+^ concentration in the filtrate with the total addition of SHMP. After 5 min stirring, the Ca^2+^ concentration in the case of no SHMP was 497.3 mg/mL, but it decreased to 186.4 mg/mL upon increasing the SHMP addition to 0.2%. The Ca^2+^ concentration became almost constant as the total addition of SHMP was increased to >0.2%. Comparison of Figure 3a,b reveals that the Ca^2+^ concentration decreased with increasing the stirring time. The above results indicated that the addition of SHMP inhibited the dissolution of Ca^2+^, which was additionally assisted by stirring, and with the increase of stirring time, the inhibitory effect is better. This was consistent with that found from the adsorption tests presented above (Section 3.1), and could be similarly explained. In the presence of SHMP, it combined with Ca^2+^ to form water-soluble [Ca_2_(PO_3_)_6_]^2−^ ions (Reaction (1)) [30,31,32], which were adsorbed onto the surfaces of cement particles and subsequently re-precipitated with them, resulting in the reduced Ca^2+^ concentration in the filtrates. The stirring promoted the chelation of SHMP with Ca^2+^ and thus the whole process stated above. According to Figure 3, the Ca^2+^ concentration reached the minimum upon addition of about 0.2% SHMP, indicating the best water reducing effect.
(NaPO_3_)_6_ + 2Ca^2+^ → [Ca_2_(PO_3_)_6_]^2−^ + 6Na^+^(1)

### 3.3. ζ-Potential of Cement Particles

Figure 4 shows the ζ-potential of cement particles as a function of the addition amount of SHMP. ζ-potential in the case of without SHMP was 8.5 mV. However, it changed to −14.8 and −20.2 mV upon adding, respectively, 0.1% and 0.2% SHMP. Upon further increasing the SHMP addition from 0.2% to 0.4%, the ζ-potential almost did not change. These results and their significance can be discussed as follows.

As indicated by Reaction (1) and mentioned above, when SHMP was added, water soluble [Ca_2_(PO_3_)_6_]^2−^ ions were formed due to its chelation with Ca^2+^, which were subsequently adsorbed onto the surfaces of cement particles, resulting in a reduced concentration of P element in the filtrate. The adsorption of [Ca_2_(PO_3_)_6_]^2−^ ions on the surfaces of cement particles led to in the change of ζ-potential from positive to negative. With increasing the amount of SHMP from 0 to 0.2%, more and more [Ca_2_(PO_3_)_6_]^2−^ ions were formed and accumulated onto the surfaces of cement particles, which led to the significant increase in the absolute value of ζ-potential. When the SHMP addition was >0.2%, the surfaces of cement particles became “saturated” with complexes [Ca_2_(PO_3_)_6_]^2−^ ions, and the negative electric strength of cement particles reached the maximum value; thus, the electrostatic repulsion between the cement particles became the largest. Consequently, further increasing the amount of SHMP to above 0.2% did not lead to any obvious change in the ζ-potential value. The great increase in the ζ-potential value with the SHMP addition implied that the dispersion of cement particles could be significantly improved. When the concentration of SHMP is greater than 0.2%, the dispersibility of PCAC is not further improved. SHMP contains sodium ions, so when the concentration of SHMP is increased, the concentration of Na^+^ in the solution will increase accordingly. Keita Irisawa and others mentioned that, in the formulation of refractory castables, the main refractory raw materials, cement and sodium salt additives all contain soluble sodium, which is easy to form into low-melting sodium salt, which is easy to collapse at high temperatures, which affects the refractoriness of the castable and other high-temperature performance properties [24]. The purpose of this research is to improve the performance of PCAC by adding SHMP, so that it can better act as a binder in the castable. Based on Figure 3 and that discussed above, the optimal addition of SHMP for achieving the best dispersion effect was around 0.2%. The excessive addition of SHMP beyond 0.2% would not make further improvement in the dispersibility of PCAC. On the contrary, due to the increase in the concentration of Na^+^, it will ultimately affect the refractoriness and other high-temperature properties of castable products.

### 3.4. Rheological Properties

Figure 5 demonstrates the relationship between shear stress and shear rate in the cases of cement-water slurries added with different amounts of SHMP [33]. According to the rheological theory, all the slurries belonged to non-Newtonian fluids. As can be seen from Figure 5, the shear stress generally increased with increasing the shear rate. In the case of without SHMP, the shear stress decreased rapidly as the shear rate increased from 0.1 to 16 s^−1^, which was probably caused by the destruction of the flocculated structures due to the rotor rotation. Furthermore, at a given shear rate, the shear stress generally decreased with increasing the addition amount of SHMP, suggesting that SHMP was very effective in r reducing/avoiding the agglomeration of fine particles and the generation of flocculation structure.

Figure 6 further displays viscosity values of cement-water slurries added with different amounts of SHMP, versus shear rates. The viscosity in general decreased with increasing the shear rate. Furthermore, at a given shear rate, it decreased with increasing the addition amount of SHMP, showing the shear-thinning behavior in most cases (except for in the case of addition of 0.4% SHMP, where the viscosity, changed very little with increasing the shear rate). The decrease in viscosity could be attributed to the destruction of the flocculated structures. The results shown in Figure 5 and Figure 6 indicated that the addition of appropriate amounts of SHMP (around 0.2% in this work) could make a significant improvement in the rheological properties of cement-water systems, which is believed to be beneficial to the improvements in rheological properties and workability of a PCAC-bonded castable.

### 3.5. Effect of SHMP Addition on Hydration Behavior of Calcium Aluminate Cement

As is well documented, the hydration of calcium aluminate cement involves three main steps: dissolution, nucleation and setting. Upon combination with water, Ca^2+^ and Al(OH)${}_{4}^{-}$ are released from the cement. When their concentrations reach a critical level, hydration products will start to nucleate and then precipitate [34,35]. These hydration steps are closely related to temperature, CAH_10_ is the main hydration products at temperatures less than 20 °C and C_2_AH_8_ and AH_3_ are the main hydration products above 20 °C; however, at a temperature above 35 °C (the test temperature in this work was at about 37 °C), CA and CA_2_ in the cement react with water according to Reactions (2) and (3) [35,36,37].
3CA + 12H → C_3_AH_6_ + 2AH_3_(2)
3CA_2_ + 21H → C_3_AH_6_ + 5AH_3_(3)

However, these hydration reactions were significantly affected by the addition of SHMP, as will be described and discussed below.

Figure 7 illustrates the effect of SHMP addition on the hydration extent of PCAC (after 1 d hydration), revealing that 3CaO·Al_2_O_3_·6H_2_O (C_3_AH_6_) was formed as the main hydration product, and its diffraction peaks in the sample without SHMP (Figure 5a) were higher than in the samples added with different amounts of SHMP (Figure 7b–e). In the latter, Al_4_(PO_4_)_3_(OH)_3_·9H_2_O (Vantasselite) was also detected [38], and it decreased with increasing the addition amount of SHMP. The results of XRD can only judge the type of substance, not the specific quantity, and the integrated area of the peak line can represent the relative content. In several samples, except for the different concentration of SHMP, other conditions are the same. We compared the peak-line integrated areas of C_3_AH_6_, CA, CA_2_ and other substances in different samples to determine the relative change of their contents in different samples. With increasing the addition amount of SHMP, C_3_AH_6_ decreased whereas both CA and CA_2_ increased. Figure 8 further gives SEM images of the samples after 1 d hydration. Granular C_3_AH_6_ was formed in the hydrated microstructure, but its amount decreased after increasing the addition amount of SHMP, especially when the addition was ≥0.1%. Combination of the results shown in Figure 7 and Figure 8 reveals that the addition of SHMP inhibited the hydration processes of CA and CA_2_. This can be explained as follows:

As mentioned earlier in this paper, [Ca_2_(PO_3_)_6_]^2−^ ions formed by the chelation of SHMP with Ca^2+^ were adsorbed onto the surfaces of cement particles, which reduced the contact area between the cement particles and water, leading to the significant delay in the hydration of CA and CA_2_. Moreover, the nucleation and precipitation of hydration product requires sufficiently high concentrations of Ca^2+^ and Al(OH)${}_{4}^{-}$ [14]. However, this was not the case when SHMP was added. As discussed above, the Ca^2+^ concentration was very low when SHMP was present. Therefore, the nucleation and precipitation of hydration products would be inhibited. For the above reasons, when SHMP was used as a dispersing agent for calcium aluminate cement, its addition amount should be carefully controlled. Otherwise, its excessive addition could lead to an extended hydration time of calcium aluminate cement, and significant degradation in the high-temperature properties.

## 4. Conclusions

(1) Complexes [Ca_2_(PO_3_)_6_]^2−^ ions initially formed from the reaction of sodium hexametaphosphate (SHMP) with Ca^2+^ ions were adsorbed onto the surfaces of cement particles. When the addition amount of SHMP was 0.05%, the adsorption ratio reached 99%. However, when the addition amount was increased to 0.4%, the adsorption ratio adversely decreased to 89%.

(2) The adsorption of complex [Ca_2_(PO_3_)_6_]^2−^ ions onto the surfaces of cement particles, in the case of SHMP addition, could change their electrochemical properties and improve the rheological properties and workability of a cement-bonded castable. When the addition of SHMP was less than 0.2%, the absolute value of ζ-potential increased with the addition amount of SHMP. However, it became almost constant upon increasing the addition of SHMP to >0.2%, suggesting that the optimal addition of SHMP was about 0.2%.

(3) The complexes [Ca_2_(PO_3_)_6_]^2−^ ions adsorbed onto the surfaces of cement particles could inhibit the dissolution of Ca^2+^ ions, and retard the hydration of CA and CA_2_, resulting in an extended hydration time of cement particles.

## Figures and Tables

**Figure 1 materials-13-05229-f001:**
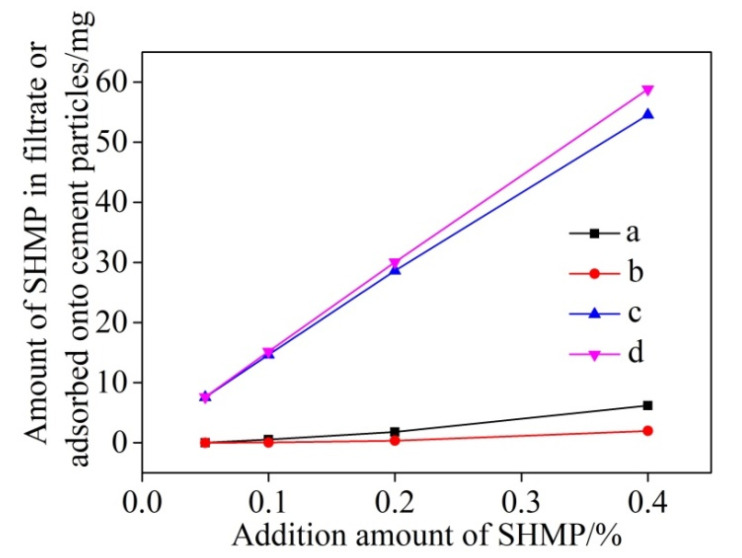
Amounts of SHMP in the filtrate after respectively: 5 min (a) and 30 min (b) stirring, and those adsorbed onto cement particles after respectively: 5 min (c) and 30 min (d) stirring.

**Figure 2 materials-13-05229-f002:**
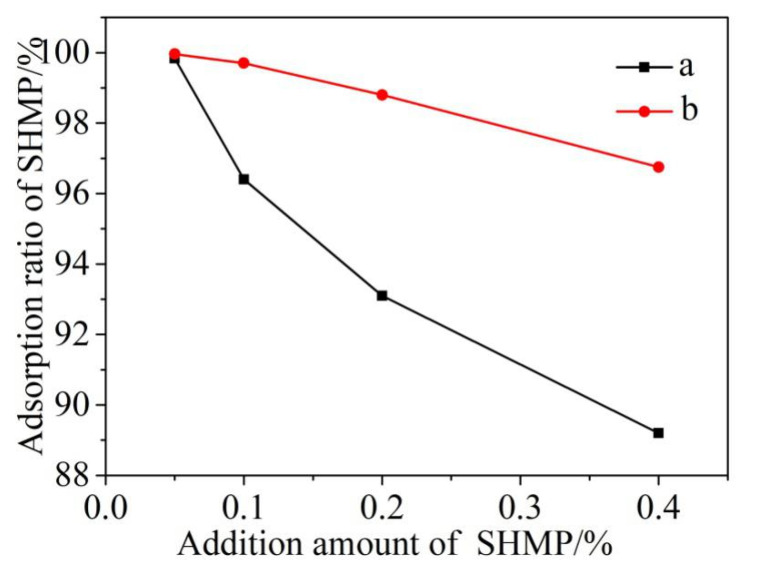
Adsorption ratio of SHMP after 5 min (a) and 30 min (b) stirring, respectively, as a function of the addition amount of SHMP.

**Figure 3 materials-13-05229-f003:**
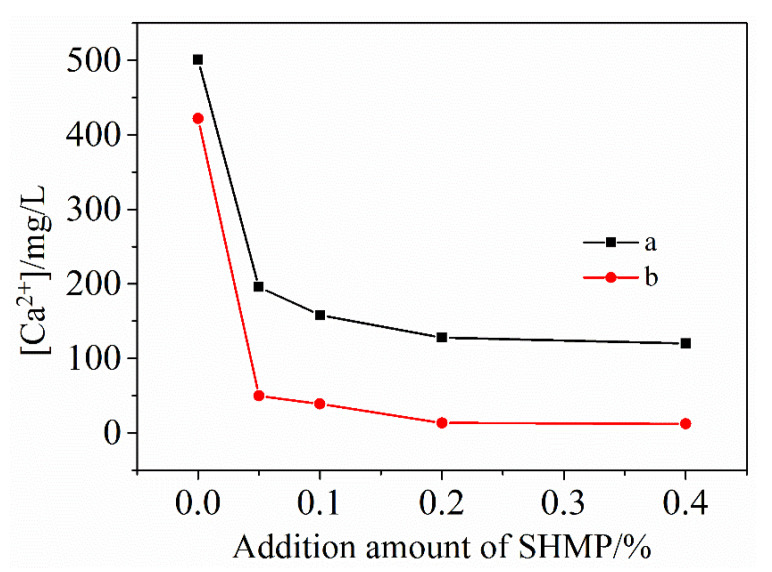
Effect of SHMP addition on Ca^2+^ concentration in the filtrate after stirring for respectively: 5 min (a) and 30 min (b).

**Figure 4 materials-13-05229-f004:**
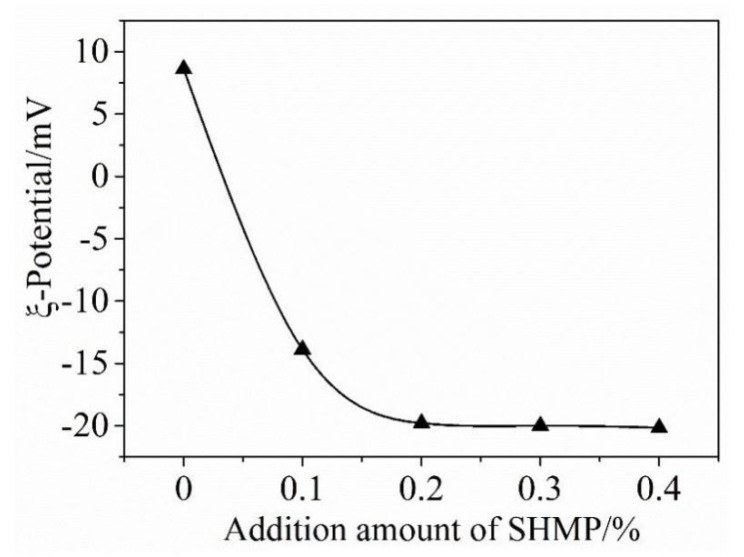
ζ-potential of cement particles as a function of addition amount of SHMP.

**Figure 5 materials-13-05229-f005:**
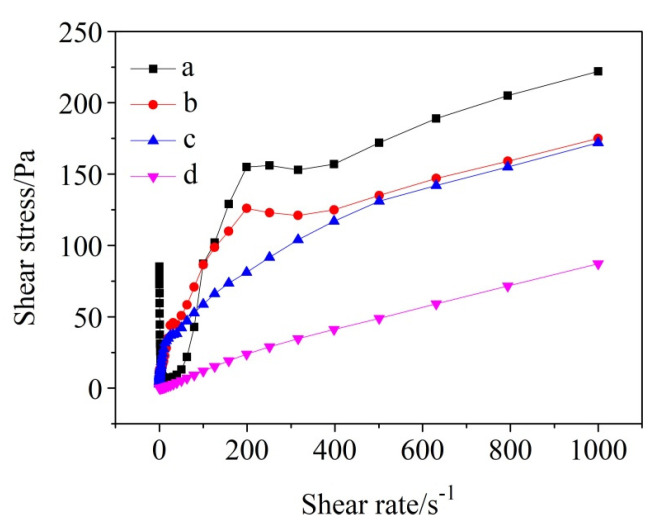
Shear stress versus shear rate curves for cement-water slurries added with different amounts of SHMP: (a) without SHMP, (b) with 0.1% SHMP, (c) with 0.2% SHMP, and (d) with 0.4% SHMP.

**Figure 6 materials-13-05229-f006:**
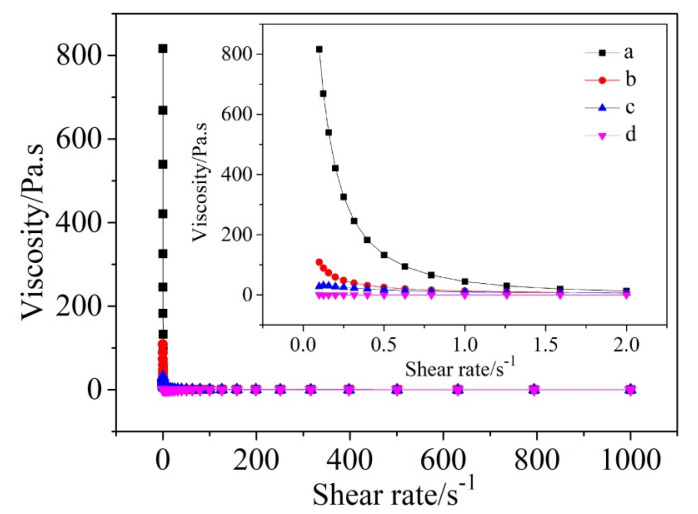
Viscosity–shear rate curves for cement-water slurries added with different amounts of SHMP: (a) without SHMP, (b) with 0.1% SHMP, (c) with 0.2% SHMP, and (d) with 0.4% SHMP.

**Figure 7 materials-13-05229-f007:**
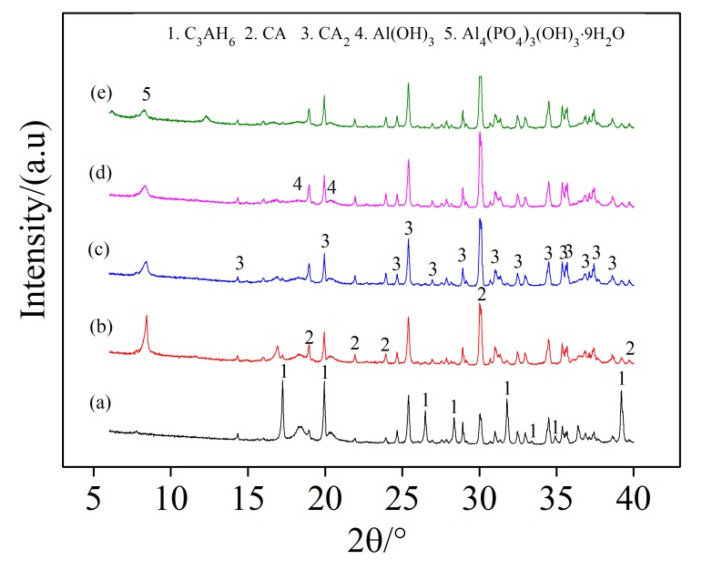
X-ray diffraction (XRD) patterns of PCAC after 1 d hydration: (a) without SHMP, (b) with 0.05% SHMP, (c) with 0.1% SHMP, (d) with 0.2% SHMP, and (e) with 0.4% SHMP.

**Figure 8 materials-13-05229-f008:**
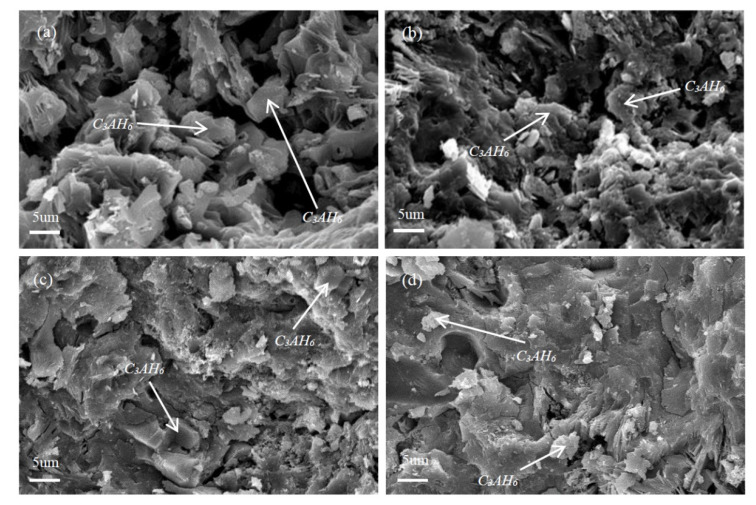
Scanning electron microscopy (SEM) images of PCAC after 1 d hydration: (**a**) without SHMP, (**b**) with 0.05% SHMP, (**c**) with 0.1% SHMP, and (**d**) with 0.2% SHMP.

**Table 1 materials-13-05229-t001:** Chemical composition of PCAC.

	Al_2_O_3_	CaO	SiO_2_	Fe_2_O_3_	MgO	TiO_2_	SO_3_
w/%	68.7	28.5	0.4	0.2	0.25	0.2	0.15

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
