# Peer review of "Effects of Sodium Hexametaphosphate Addition on the Dispersion and Hydration of Pure Calcium Aluminate Cement"

_materials, 2020, doi:10.3390/ma13225229_

Round 1
Reviewer 1 Report
In this study, the effect of sodium hexametaphosphate (SHMP) on the hydration and dispersion of calcium aluminate cement was investigated. However, SHMP is not a construction material. Generally, superplasticizers (especially, polycarboxylate-ether type) have been used satisfactorily over the past decades for the same purpose.
Overall, the manuscript has not been faithfully written. In particular, it is difficult to find the motivation and justification of this study. Also, the literature review has not been sufficiently performed. Many of the cited references are written in Chinese. It should be replaced with more influential and relevant papers.
Most crucially, the experimental results are very insufficient to claim the conclusion of the study. Therefore, additional experiments should be performed, and moreover in-depth discussion are required based on the experimental evidence.
Author Response
Point 1: In this study, the effect of sodium hexametaphosphate (SHMP) on the hydration and dispersion of calcium aluminate cement was investigated. However, SHMP is not a construction material. Generally, superplasticizers (especially, polycarboxylate-ether type) have been used satisfactorily over the past decades for the same purpose.
Response 1: As you mentioned, polycarboxylate ether superplasticizer is a kind of high performance water reducer and a kind of cement dispersant in the application of cement concrete, widely used in highway, bridge, dam, tunnel, high-rise building and other projects. SHMP( Sodium hexametaphosphate ) is often used as water reducing agent in the castable combined with PCAC( pure calcium aluminate cement ) which include silica fume content of 2-8%, but the water reducing mechanism of SHMP superplasticizer on PCAC is rarely studied.
Calcium aluminate cement is used as the binder of unshaped refractory castable , and it is not very suitable to use polycarboxylate ether as the superplasticizer of PCAC ( pure calcium aluminate cement ) with silica fume content of 2-8%, and SHMP is one of the most effective dispersants. The purpose of this paper is to explore its working mechanism and the optimal amount of SHMP through experiments. We've modified the introduction to make it look more reasonable, as you can see on lines 55-62.
Point 2: Overall, the manuscript has not been faithfully written. In particular, it is difficult to find the motivation and justification of this study.
Response 2: As for the motivation and reason of this paper, we have mentioned in the introduction that among the binders for refractory castables, pure calcium aluminate cement (PCAC) with CaO · Al2O3 (CA) and CaO · 2Al2O3 (CA2) as the main phase is commonly used . Due to the low overall mechanical strength of PCAC-bonded castable , dispersants are usually added to ensure good fluidity and harden ability , and further improving other properties and overall properties of the castable. Sodium hexametaphosphate (SHMP) is one of the most effective dispersants. However, the research on the dispersion / retarding effect of SHMP on pure calcium aluminate cement is still limited, so the mechanism of its action is not fully understood. To address this, in the present work, ζ-potential, adsorption amount of SHMP, the concentrations of P element and Ca2+ and rheological properties corresponding to different SHMP additions were examined, based on which, and the hydration behavior of SHMP investigated, the relevant mechanism underpinnings was discussed. The work would provide an insight into the effects of SHMP on the dispersion and water reduction in a PCAC bonded castable system, and find the optimal addition amount of SHMP.
Point 3: The literature review has not been sufficiently performed. Many of the cited references are written in Chinese. It should be replaced with more influential and relevant papers.
Response 3: In terms of literature review and reference citation, we would like to thank the reviewers for their suggestions, which are of great value to the revision and improvement of our paper, and also of great significance to the research of our article. We refer to a large number of papers published in well-known journals at home and abroad related to this study carefully, and rewrite the introduction part, hoping to be approved.
Point 4: Most crucially, the experimental results are very insufficient to claim the conclusion of the study. Therefore, additional experiments should be performed, and moreover in-depth discussions are required based on the experimental evidence.
Response 4: Our conclusions are based on the experimental results and the research of many related scholars. Thank the reviewers for their rigorous work and inquiries on our manuscript. Therefore, we have revised the relevant parts (The abstract (line18-22), the third part( line127-130、line192-199 ), etc.) to make the structure and content of our manuscript more reasonable. Thank you again for your comments on our article, it is of great significance to improve our manuscript.
Finally, thank you for your helpful comments. We believe that with the help of your advice, our manuscript has been greatly improved.
Reviewer 2 Report
This paper investigated the effects of sodium hexametaphosphate on the dispersion and hydration of pure calcium aluminate cement based on the experimental data. This is well written and easy to follow. The results are interesting. However, to address the following questions, it can be further improved.
- What is the reason to select the water-to-cement ratio used in the experiment?
- In p.5, the authors wrote, “the optimal addition of SHMP for achieving the best dispersion effect was 0.2%. The excessive addition of SHMP beyond 0.2% would not make further improvement in the dispersibility of PCAC, Instead, would affect negatively the refractoriness and other high temperature properties of the final castable products due to the formation of too much low melting phases from Na+.” Also, in the conclusion (2), the authors claimed that the optimal addition of SHMP was about 0.2%. However, the authors did not provide any proof for the upper limit of SHMP addition in terms of the resistance to high temperature at all. Thus, the “optimal addition of SHMP” cannot be determined from the experimental data presented in this paper and the sentence “the excessive addition of SHMP beyond 0.2% would affect negatively the refractoriness and other high temperature properties of the final castable products” is not appropriate.
- Check for the format of the graphs in editorial regulation of the journal.
- Typo error: line 126 in p.4, “couls be” should be “could be.”
Author Response
Point 1: What is the reason to select the water-to-cement ratio used in the experiment?
Response 1: Water-to-cement ratio refers to the weight ratio of water addition and cement dosage in concrete. The water cement ratio affects the rheological properties of concrete, the condensed structure of cement slurry and the compactness after hardening. Therefore, under the given composition materials, the water-to-cement ratio is the main parameter to determine the strength, durability and a series of other physical and mechanical properties of concrete. If the water cement ratio is too small, the hydration heat will be large, and the concrete is easy to crack, and the workability of concrete is poor, which is not conducive to the field construction operation. If the water cement ratio is too high, the concrete strength will be reduced. Water reducing agent can reduce the proportion of water in the slurry without affecting its normal fluidity. Water reducing agent is usually used to control the water cement ratio in engineering, so as to meet the application requirements. We have improved and supplemented lines 44-50 in the introduction, and explained the working mechanism of the water reducer in detail.
Point 2: In p.5, the authors wrote, “the optimal addition of SHMP for achieving the best dispersion effect was 0.2%. The excessive addition of SHMP beyond 0.2% would not make further improvement in the dispersibility of PCAC, Instead, would affect negatively the refractoriness and other high temperature properties of the final castable products due to the formation of too much low melting phases from Na+.” Also, in the conclusion (2), the authors claimed that the optimal addition of SHMP was about 0.2%. However, the authors did not provide any proof for the upper limit of SHMP addition in terms of the resistance to high temperature at all. Thus, the “optimal addition of SHMP” cannot be determined from the experimental data presented in this paper and the sentence “the excessive addition of SHMP beyond 0.2% would affect negatively the refractoriness and other high temperature properties of the final castable products” is not appropriate.
Response 2: In reference [33], Keita irisawa et al. found that the presence of Na+ in the castable is easy to form soluble sodium salt and form low melting point substance, which is easy to collapse at high temperature. In the castables formula, the sources of soluble sodium salt include refractory raw materials, cement and sodium salt additives. In this experiment, when the concentration of SHMP increases from 0% to 0.2%, the absolute value of cement particle potential increases gradually, indicating that the dispersion of cement particles is significantly improved. When the concentration of SHMP exceeds 0.2%, there is no obvious change in cement particle potential, which indicates that increasing the concentration of SHMP do not further improve the dispersion of PCAC. The purpose of adding SHMP to calcium aluminate cement is to improve the performance of cement and make it play a better role as a binder in the castables. However, the SHMP contains Na+. Therefore, when the concentration of SHMP exceeds 0.2%, not only the hydration process of cement can not be improved, but also the content of Na+ will be increased, which is not conducive to the fire resistance and other high-temperature properties of the castables. Thank the reviewers for their conscientiousness and responsibility. We have supplemented and improved lines 193-201 to make the description more rigorous.
Point 3: Check for the format of the graphs in editorial regulation of the journal.
Response 3: Thanks for the reminder of the reviewer, we have revised the editing format of the graphs in the paper ( line 151-152、154、172-173).
Point 4: Typo error: line 126 in p.4, “couls be” should be “could be.”
Response 4: Thanks to the reviewers for their rigorous attitude, that word is indeed "could", and we have revised it.
We appreciate for Editors and Reviewers’ warm work earnestly, and hope that the corrections will meet with your approval.
Once again, thank you very much for your comments and suggestions.
Reviewer 3 Report
This work provides some interesting information on the role of SHMP on the hydration of CAC systems. However there are some concerns with the methods used and analysis in the paper that should be addressed.
- The authors reference list and literature review is quite sparse and ignores much of the basic underpinning of what CAC is and how it hydrates. Given that this paper is looking at hydration kinetics of CAC, it is important to include that information.
- Line 75. The authors note that they are using zeta potential measurements to show how SHMP may change the surface charge of particles and thus improve dispersion. However, the authors used the SHMP and cement mixture in very low concentrations. Zeta potential of mineral structures is known to be impacted by the pH of a system. This diffuse system would most likely not have achieved the pH that would be standard for a CAC system (11-12). Since the authors did not report the pH, its not actually known. Authors need to provide the pH of the solutions they tested, and if it is lower than a normal CAC system, discuss how that may impact the results.
- Line 82. The authors need to discuss clearly in the procedures how temperature of the samples was controlled and at which temperature all the workd was done, and report the temperature of the specimens. Hydration kinetics of CAC is extremely temperature dependent. This temperature dependency should also be discussed by the authors.
- Figure 1 and analysis thereof. The authors state that because more SHMP was adsorbed after 30 minutes instead of after 5 minutes, that stirring helped the adsorption. However, the results do not show this. The results show that in a stirred sample, more SHMP adsorbed after additional time. The time was the only thing that changed. Without showing the results of a system that wasn't stirred, the authors cannot remark on whether stirring was important or not.
- Line 192: Given results by Scrivener's group, particularly research from C. Gosselin, it is likely that the metastable products actuall form first, at this temperature, and then there is quick conversion fo C3AH6.
- line 197. The way the authors present this, it seems like they are saying that CA and CA2 formed in place of C3AH6 with increasing amounts of SHMP. However, it seems that this is likely a retarding effect. The authors most likely just need to rewrite this to make that more clear.
- Figure 7 and commentary thereof. The authors state that from the XRD results that they could determine there was more of one component than the other between systems. However, a basic XRD analysis can't really tell if you if there is more or less. Changes in peak intensity are not necessarily related to amount unless you do at least semi-quantitative analysis or Rietveld analysis of XRD. With what is presented, the authors can really just state what is present in each system, not how much.
Author Response
Point 1: The authors reference list and literature review is quite sparse and ignores much of the basic underpinning of what CAC is and how it hydrates. Given that this paper is looking at hydration kinetics of CAC, it is important to include that information.
Response 1: Thank you very much for your suggestion, which is very valuable for the improvement of our manuscript. The hydration process of PCAC is described in detail in lines 236-243 of the manuscript. At the same time, we improved and supplemented lines 43-54 in introduction, and introduced the water reducing mechanism of water reducing agent on cement.
Point 2: Line 75. The authors note that they are using zeta potential measurements to show how SHMP may change the surface charge of particles and thus improve dispersion. However, the authors used the SHMP and cement mixture in very low concentrations. Zeta potential of mineral structures is known to be impacted by the pH of a system. This diffuse system would most likely not have achieved the pH that would be standard for a CAC system (11-12). Since the authors did not report the pH, its not actually known. Authors need to provide the pH of the solutions they tested, and if it is lower than a normal CAC system, discuss how that may impact the results.
Response 2: During the experiment, the pH value we measured was around 11.2, decreasing slightly with the test time increasing,reaching the standard PH value (11-12) in the CAC system.Thank you for your reminder, we have made improvements in lines 107-108.
Point 3: Line 82. The authors need to discuss clearly in the procedures how temperature of the samples was controlled and at which temperature all the workd was done, and report the temperature of the specimens. Hydration kinetics of CAC is extremely temperature dependent. This temperature dependency should also be discussed by the authors.
Response 3: When measuring cement rheological properties and potential experiments, the time of the experiment process is relatively short, and we all carry out it at room temperature in summer (the temperature is around 37℃). When studying cement hydration, first prepare the mold, then put it in a 37℃ incubator for natural curing for 24 hours, and then demold. X-ray diffraction (XRD) and scanning electron microscope (SEM) are used to analyze the phase transition of the sample and observe the microstructure of the sample. We have made supplementary explanations in lines 114 and 239-240.
Point 4: Figure 1 and analysis thereof. The authors state that because more SHMP was adsorbed after 30 minutes instead of after 5 minutes, that stirring helped the adsorption. However, the results do not show this. The results show that in a stirred sample, more SHMP adsorbed after additional time. The time was the only thing that changed. Without showing the results of a system that wasn't stirred, the authors cannot remark on whether stirring was important or not.
Response 4: At the beginning of the test, the cement particles sink to the bottom, and the amount of SHMP adsorbed on the surface is very small, almost no. After standing for 5 min and 30 min, the concentration of p element was measured and compared, and it was found that there was no change. Therefore, stirring is used to increase its adsorption capacity. We have included this comparison in lines 128-131 of the manuscript.
Point 5: Line 192: Given results by Scrivener's group, particularly research from C. Gosselin, it is likely that the metastable products actuall form first, at this temperature, and then there is quick conversion fo C3AH6.
Response 5: As you mentioned, calcium aluminate cement could dissolve in water, and this process is a metastable state. When the temperature is higher than 35℃, it will quickly react and transform into C3AH6, It is described in lines 238-242 of the manuscript. What we are studying is that after adding SHMP, it will affect the hydration process of PCAC and determine the optimal amount of SHMP.
Point 6: Line 197. The way the authors present this, it seems like they are saying that CA and CA2 formed in place of C3AH6 with increasing amounts of SHMP. However, it seems that this is likely a retarding effect. The authors most likely just need to rewrite this to make that more clear.
Response 6: Thank you very much for your helpful comments. As you said, the addition of SHMP delayed the processes of CA and CA2 to generate C3AH6.There may be a problem with our expression. We have made corrections in the manuscript.
Point 7: Figure 7 and commentary thereof. The authors state that from the XRD results that they could determine there was more of one component than the other between systems. However, a basic XRD analysis can't really tell if you if there is more or less. Changes in peak intensity are not necessarily related to amount unless you do at least semi-quantitative analysis or Rietveld analysis of XRD. With what is presented, the authors can really just state what is present in each system, not how much.
Response 7: Your comment is correct. The results of XRD can only judge the type of the substance, but not the specific quantity, but the area formed by the peak line and horizontal line can represent the relative content. In this study, except the concentration of SHMP added in several samples is different, other conditions are the same. We can judge the relative content of the same substance in different samples by comparing the area formed by the peak line and horizontal line of the same substance in different samples, and roughly judge the relative content changes of C3AH6, CA, CA2 and other substances . Hope you can accept this explanation. We have improved and supplemented lines 251-255.
Finally, thank you for your helpful comments. We believe that with the help of your advice, our article has been greatly improved.
Round 2
Reviewer 1 Report
The manuscript has been improved, but still needs to be revised. Again, improve the manuscript by taking into account the comments below.
- The authors mentioned that ‘SHMP is often used as water reducing agent, especially for calcium aluminate cement’. However, the reviewer hard to trust this answer.
- Although previous studies have been summarized to reinforce the study's justification, most of them are irrelevant to the main topic of the study.
- The description of the interaction between superplasticizer and cement particles is insufficient. Please write it carefully.
Author Response
Response to Reviewer 1 Comments
Point 1: The authors mentioned that‘SHMP is often used as water reducing agent, especially for calcium aluminate cement’. However, the reviewer hard to trust this answer.
Response 1: Thank you for your comments. This is a mistake in our presentation. In fact, sodium hexametaphosphate is a more popular dispersant, water reducing agent, coagulant and viscosity regulator in refractory industry. Because of its long chain, it can ensure that the refractory castable can achieve the required fluidity and easy pouring construction performance, and promote the refractory casting construction lining to form a low porosity and high mechanical strength structure. The previous statement is relatively simple, which makes the reader unconvinced. We have revised the corresponding part (lines 63-72,74-80).
Point 2:Although previous studies have been summarized to reinforce the study's justification, most of them are irrelevant to the main topic of the study.
Response 2: Thank you very much for your comments. We have revised the literature review (74-80), mainly focusing on the impact of SHMP on the dispersion of materials and cement hydration, which are relevant to the main topic of this study.
Point 3:The description of the interaction between superplasticizer and cement particles is insufficient. Please write it carefully.
Response 3: Regarding the interaction between superplasticizer and cement particles, we have made improvements and supplements in lines 42-56.